# Inducible Enrichment of Osa-miR1432 Confers Rice Bacterial Blight Resistance through Suppressing *OsCaML2*

**DOI:** 10.3390/ijms222111367

**Published:** 2021-10-21

**Authors:** Yanfeng Jia, Quanlin Li, Yuying Li, Wenxue Zhai, Guanghuai Jiang, Chunrong Li

**Affiliations:** Institute of Genetics and Developmental Biology, Chinese Academy of Sciences, Beijing 100101, China; yfjia@genetics.ac.cn (Y.J.); quanlinli@genetics.ac.cn (Q.L.); lyear@genetics.ac.cn (Y.L.); wxzhai@genetics.ac.cn (W.Z.)

**Keywords:** rice bacterial blight, osa-miR1432, *OsCaML2*

## Abstract

MicroRNAs (miRNAs) handle immune response to pathogens by adjusting the function of target genes in plants. However, the experimentally documented miRNA/target modules implicated in the interplay between rice and *Xanthomonas oryzae* pv. *oryzae* (*Xoo*) are still in the early stages. Herein, the expression of osa-miR1432 was induced in resistant genotype IRBB5, but not susceptible genotype IR24, under *Xoo* strain PXO86 attack. Overexpressed osa-miR1432 heightened rice disease resistance to *Xoo*, indicated by enhancive enrichment of defense marker genes, raised reactive oxygen species (ROS) levels, repressed bacterial growth and shortened leaf lesion length, whilst the disruptive accumulation of osa-miR1432 accelerated rice susceptibility to *Xoo* infection. Noticeably, *OsCaML2* (*LOC_Os03g59770*) was experimentally confirmed as a target gene of osa-miR1432, and the overexpressing *OsCaML2* transgenic plants exhibited compromised resistance to *Xoo* infestation. Our results indicate that osa-miR1432 and *OsCaML2* were differently responsive to *Xoo* invasion at the transcriptional level and fine-tune rice resistance to *Xoo* infection, which may be referable in resistance gene discovery and valuable in the pursuit of improving *Xoo* resistance in rice breeding.

## 1. Introduction

In plants, microRNAs (miRNAs) manipulate various biological processes, including developmental plasticity, stress responses and symbiotic/parasitic interactions by regulating target gene expression [1]. With plants predominantly dependent on resistance protein-coding genes to execute immunity against invading pathogen, miRNAs may function as crucial regulators in plant immune responses. The current knowledge of miRNA/target modules in plant–pathogen interaction are mainly concentrated on the pathosystems of plant and fungi or viruses, and only small-scale reports were involved in plant and bacteria. Therefore, to enrich the evidence of miRNA/target modules on plant bacterial disease would contribute to the overall mechanism dissection of plant–pathogen interaction and disease management, especially in crop resistance breeding.

Rice (*Oryza sativa* L.) offers calorie intake for more than half of the world’s population, the worldwide production of which is restricted by the bacterial pathogen *Xanthomonas oryzae* pv. *oryzae* (*Xoo*) [2,3,4]. Multivarious conventional measures, including germicidal chemicals, biological agents and resistance breeding, are widely employed to control *Xoo* infection in rice cultivation. Unfortunately, the extensive use of germicidal chemicals brings environmental pollution and the tachytelic evolution of drug-resistant *Xoo* populations [5,6], and biological agents are not always effective. The deployment of cultivars carrying resistance gene, the most economical, environmentally friendly and effective method, is obstructed because of resistance loss and the emergence of new *Xoo* pathotypes [7,8,9]. Currently, an effective strategy for maintaining rice resistance to *Xoo* pathotypes is to excavate new resistance genes or reprogram the responsive patterns of TALE-triggered susceptibility genes, which is separately achieved by genome-wide association study (GWAS) approaches [10,11] and Cas9-mediated cutting events [12,13]. With the exception of the GWAS and TALE-dependent resistance gene mining methods, miRNA-targeted genes may provide another access to mine resistance-associated genes.

Although abundant candidate targets of miRNAs have been monitored via the bioinformatic crosstab survey of small RNA sequencing data and transcriptome or degradome data in *Xoo*-challenged rice leaves [14,15], empirical investigations on the functionality characterization of miRNA/target modules in rice-*Xoo* interaction are slowly releasing. Osa-miR1861k may render rice immune to *Xoo* infection by targeting *LOC_Os03g18710* and *LOC_Os06g44970* [16]. Conversely, osa-miR169o inhibits the accumulation of the *NF-YA4* gene, which impedes *PR* and *NRT2* transcription and increases susceptibility to *Xoo* in rice [17]. The aforementioned targets of miRNAs were presented indirectly for rice defense against *Xoo*. Importantly, the transcript enrichment of *OsIPA1**,* one target of osa-miR156, weakens GA-mediated *Xoo* susceptibility by stabilizing OsSLR1 [18], and the quantitative resistance of the *OsNBS8R* gene to *Xoo* was attenuated by non-TAL effector XopQ-inducible osa-miR1876 through DNA methylation [19], which were direct proofs of miRNA/target modules functioning in rice-*Xoo* interaction. 

Pioneer reports uncovered that osa-miR1432 boosts the filling rate by targeting *OsACOT* [20] and may be responsive to *Xoo* stimulus in rice [15]. In this study, we found that osa-miR1432 is induced specifically by the *Xoo* strain PXO86 in rice-resistant genotype IRBB5 and positively regulates rice resistance against *Xoo.* Furthermore, we experimentally demonstrated that *LOC_Os03g59770* (*OsCaML2*) is a genuine target of osa-miR1432, and *OsCaML2* overexpressors develop disease susceptibility to *Xoo*.

## 2. Results

### 2.1. Osa-miR1432 Expression Is Induced in Resistant Genotypes during Xoo Invasion

In our previous small RNA sequencing data, osa-miR1432 was differentially responsive to *Xoo* strain PXO86 invasion in the susceptible genotype IR24 and resistant genotype IRBB5 [16]. To confirm this, reverse transcription quantitative PCR (RT-qPCR) assays were executed to depict the responsive pattern of osa-miR1432 at the early stage of *Xoo*-challenged IR24 and IRBB5 (Figure 1A). In contrast to mock treatments, *Xoo*-responsive expression of osa-miR1432 showed no obvious difference in infected IR24 leaves except for 24 and 48 h post inoculation (hpi), with reduced accumulation. However, a higher enrichment of osa-miR1432 was observed from 8 hpi to 96 hpi in *Xoo*-treated IRBB5 than in mock-treated plants (Figure 1B), indicating that osa-miR1432 is specifically induced by *Xoo* strain PXO86 invasion in IRBB5. These results suggest that osa-miR1432 may engage in rice immunity against *Xoo* invasion.

### 2.2. Overexpressing Osa-MiR1432 Imparts Rice Resistance to Xoo Infection

To verify whether osa-miR1432 mediates disease resistance to *Xoo* infection in rice, overexpressing osa-miR1432 (OE1432) transgenic plants were generated under the Taipei 309 (TP309) background. Out of 32 independent transgenic lines, two lines with high expression levels of osa-miR1432 were chosen to evaluate the disease reaction (Figure 2A). The leaf disease lesions caused by *Xoo* strain PXO86 in transgenic plants OE1432-1 and OE1432-7 were shorter than those of the control TP309 (Figure 2B,C), implying that osa-miR1432 may operate as a positive regulator in rice resistance to *Xoo* infection. Subsequently, the leaf lesions of OE1432 lines suffering the effects of *Xoo* strains PXO112, PXO99 and PXO339 were measured and weaker relative to TP309 (Appendix A). These results suggest the universality of overexpressing osa-miR1432 in strengthening rice immunity to *Xoo* infection.

In order to determine how osa-miR1432 blocks the pathogenicity of *Xoo* in rice, the typical immune response indicators were analyzed, including the initial expression of defense marker genes, the production of reactive oxygen species (ROS) and the bacterial growth curve. Firstly, the intrinsic expression levels of the defense marker genes, *OsPBZ1* (*Oryza sativa* pathogenesis-related protein 10a) [21], *OsPR1a* (pathogenesis-related protein 1a) and *OsPR1b* (pathogenesis-related protein 1b) [22], in OE1432 plants were examined and superior to their counterparts in TP309 plants (Figure 2E), suggesting that OE1432 plants may possess a faster immune response. Secondly, O_2_^−^ and H_2_O_2,_ the main form of ROS, were found at higher levels at 4 days post inoculation (dpi) in OE1432 plants compared with TP309 plants (Figure 2F), suggesting that OE1432 plants may employ more severe cell damage to execute disease resistance. Finally, the growth rates of *Xoo* strain PXO86 were lower in OE1432 transgenic rice plants than those in TP309 plants, ranging from 4 dpi to 20 dpi (Figure 2D), implying that osa-miR1432-mediated resistance may act in the inhibition of *Xoo* strain multiplication. These data indicate that overexpressing osa-miR1432 reinforces rice resistance against *Xoo* strain PXO86 through the elevated expression of defense marker genes, ROS production and the decreased cell number of *Xoo* strain PXO86.

### 2.3. Osa-MiR1432 Knockdown Plants Exhibit Aggravated Susceptibility to Xoo Infection

To further notarize that osa-miR1432-mediated rice immunity depends on its inducible expression, we generated endogenous osa-miR1432 knockdown (MIM1432) lines using short tandem target mimic technology [23,24] under the TP309 background. Two lines with notably reduced accumulation of osa-miR1432 were screened from 21 independent transgenic plants for disease assay (Figure 3A). By inoculating the *Xoo* strain PXO86, MIM1432-3 and MIM1432-8 lines presented more severe lesions, ranging from 13.0 cm to 13.3 cm, than TP309, 11.0 cm, in the infected leaves (Figure 3B,C). Consistent with the susceptible phenotype, the inherent transcripts of *OsPBZ1*, *OsPR1a* and *OsPR1b* were repressive in MIM1432 lines (Figure 3E). Similarly, responsive ROS levels for *Xoo* infection were low, at 4 dpi, in MIM1432 plants versus TP309 (Figure 3F). Bacterial growth survey results reflected that the PXO86 cells on the leaves of MIM1432 plants were one order of magnitude higher than those in TP309 plants after the middle stage of infection (Figure 3D). Additionally, the MIM1432 plants recapitulated the susceptibility under PXO112, PXO99 and PXO339 strain treatments (Appendix A). These results demonstrate that interrupted osa-miR1432 weakens rice resistance against *Xoo* infection.

### 2.4. Osa-miR1432 Targets the Transcripts of OsCaML2 in Plants 

Generally, disease resistance was conferred by the corresponding resistance gene in plant–pathogen interaction. To mine the downstream targets of osa-miR1432 in regulating rice immunity, eight potential targets for osa-miR1432 were predicted using the psRNATarget tool (http://plantgrn.noble.org/psRNATarget/ accessed on 22 May 2020) with default parameters (Appendix A). Coincidently, three potential targets, *LOC_Os03g59770* (*OsCaML2*), *LOC_Os04g08350* (*OsCAS*) and *LOC_Os05g07210* (*OsZIP6*), also existed in high-quality IR24-D and IRBB5-D degradome data after removing adaptor sequences and low-quality reads [15], and then were further investigated in osa-miR1432 transgenic plants. As expected, the basic mRNA levels of all three targets were signally suppressive in OE1432 lines and cumulative in MIM1432 lines (Figure 4A). Simultaneously, all these genes were differentially expressed in IR24 and IRBB5 upon infection of *Xoo* strain PXO86 (Figure 4B–D). However, compared with the mock treatment, the *Xoo*-responsive patterns of *OsCaML2* and *OsCAS* in IR24 and IRBB5 were, to some extent, opposite to osa-miR1432 in 3 dpi, whilst the expressing trend of *OsZIP6* was in keeping with osa-miR1432 (Figure 4D). Noteworthily, from the perspective of *Xoo*-dependent expression, the opposite and similar transcriptional trend, ranging from 8 hpi to 72 hpi of *OsCaML2* and *OsCAS* to osa-miR1432, appeared in *Xoo*-treated IRBB5, respectively. Thus, *OsCaML2* was speculated as the best prospective target of osa-miR1432 in rice-*Xoo* interaction.

To further affirm the regulatory relationship between *OsCaML2* and osa-miR1432, a green fluorescent protein (GFP)-based reporter assay (Appendix A) was conducted to transiently express OsCaML2-GFP or togetherwith osa-miR1432 in *Nicotiana benthamiana*. When OsCaML2-GFP and osa-miR1432 were co-expressed in *Nicotiana benthamiana*, the signal intensity of OsCaML2-GFP was gradually decreased with incremental proportion of osa-miR1432 in the injected mixture, accompanying no obvious change in GFP intensity of the control vector with or without osa-miR1432 under the same conditions (Appendix A). To test whether MIM1432 could rescue the silencing of OsCaML2-GFP guided by osa-miR1432, MIM1432 was added stepwise to the infiltrated mixture of OsCaML2-GFP and osa-miR1432, and the GFP signals became increasingly stronger (Appendix A). Taken together, the above results suggest that *OsCaML2* is an authentic target of osa-miR1432 in rice.

### 2.5. Overexpression of OsCaML2 Phenocopies the Susceptibility of MIM1432 Plants

To investigate the underlying biological function of *OsCaML2* in rice, phylogenetic tree analysis and subcellular localization were implemented. Phylogenetic tree analysis uncovered that all 38 homologues of OsCaML2 were from plants, especially in Gramineae (Appendix A). Unfortunately, the relevant biological function studies of these 38 proteins have not been reported. Concurrently, the functional compartment localization exposed that OsCaML2 was expressed in both the nucleus and cytoplasm in *Nicotiana benthamiana* cells (Appendix A), meaning OsCaML2 may participate in the signal transduction of the nucleus and cytoplasm.

To experimentally ascertain the role of *OsCaML2* in *Xoo* infection, *OsCaML2*-overexpressing (OEOsCaML2) plants, driven by UBI promoter, were created, and the disease resistance of two highly expressed lines, OEOsCaML2-5 and OEOsCaML2-6, were examined by inoculating the *Xoo* strain PXO86. The inoculated leaves from OEOsCaML2-5 and OEOsCaML2-6 lines proved to have longer lesion lengths than those of TP309 plants (Figure 5A–C), and the disease symptoms were supported by the transcript levels of defense marker genes, the histochemical detection of ROS and bacterial growth number in *OsCaML2* transgenic lines (Figure 5D–F). These results indicate that *OsCaML2* may negatively regulate the immunity of rice to *Xoo* infection.

To unearth the functional genes enacting in the immune pathway regualated by the osa-miR1432/*OsCaML2* module, the potential proteins interacting with OsCaML2 were screened from a yeast two-hybrid (Y2H) library, and two candidate proteins, OsHPGT3 and OsCSN6, were identified (Appendix A). OsCSN6 may be an iron deficiency responsive regulator to conquer Fe stress in rice [25]. The homology analysis and functional annotation suggest that OsHPGT3 may be involved in arabinogalactan metabolism of the rice cell wall, affecting the composition of the cell wall [26]. OsHPGT3 was chosen for further inspection. To determine whether the interaction between OsCaML2 and OsHPGT3 depends on the EF hand domain, the full-length OsCaML2 (1–213 aa) and two truncated variants, OsCaML2-N (containing EF hand domain at 146–174 aa) and OsCaML2-C (containing EF hand domain at 184–212 aa), were inserted into the pGBKT7 vector and co-transformed into yeast cells with pGADT7-OsHPGT3. The results showed that OsCaML2, OsCaML2-N and OsCaML2-C could interact with OsHPGT3 protein in yeast, indicating the interaction relies on the EF hand domain of OsCaML2 (Appendix A). In addition, the bimolecular fluorescence complementation (BiFC) assay showed the interaction of OsCaML2 and OsHPGT3 also existed in *Nicotiana benthamiana* (Appendix A). The above results suggest that OsHPGT3 may function as an osa-miR1432/*OsCaML2* module-mediated defense pathway.

## 3. Discussion

Recent works have disclosed that miRNA/target modules serve as an indispensable participant in rice-pathogen interaction. Despite the fact that numerous miRNA/target modules have been identified by bioinformatic prediction or omics analysis, only two regulatory modules, osa-miR156/*OsIPA1* [18] and osa-miR1876/*NBS8R* [19], were demonstrated to orchestrate rice immunity to *Xoo* utilizing the transgenic approach. In our study, overexpressing osa-miR1432 plants were shown to enhance rice resistance to *Xoo* invasion, while the endogenous blocking of osa-miR1432 avails *Xoo* multiplication in infected rice leaves. Consistently, the elevated accumulation of *OsCaML2*, the target of osa-miR1432, increases the susceptibility to *Xoo* invasion in rice. These results show that the osa-miR1432/*OsCaML2* module is implicated in the immunity response to *Xoo* invasion and it is feasible to identify resistance-associated genes by exploiting the targets of functional miRNAs.

Previous work has illustrated the regulatory role of osa-miR1432 in developing rice seeds, which negatively regulates grain size via adjusting the grain filling rate through its target, rice Acyl-CoA thioesterase (*OsACOT*) [20]. Although the similar phenotype reappeared in the genetic materials of osa-miR1432 in this study, our attention was focused on the biological function of osa-miR1432 in rice-*Xoo* interaction, namely that osa-miR1432 operates as a positive immune response regulator. *OsPBZ1, OsPR1a* and *OsPR1b,* which were induced by salicylic acid (SA), jasmonic acid (JA) and ethylene (ET), respectively [27,28,29], in OE1432 plants, were significantly higher, relative to TP309 plants, and presented contrastive results in MIM1432 plants, implying that osa-miR1432-activated resistance may be involved in an SA-JA/ET-mediated pathway (Figure 2E and Figure 3E). The *Xoo* resistance of OE1432 plants was associated with increased ROS accumulation and decreased growth rates of *Xoo* strain PXO86 during the period of rice-*Xoo* interaction (Figure 2D,F). In contrast, disturbing osa-miR1432 resulted in compromised resistance to *Xoo* invasion, which was attributable to less ROS accumulation and raised growth rates of *Xoo* strain PXO86 (Figure 3D,F).

The *Xoo* resistance endowed by overexpressing osa-miR1432 counts on the function inhibition of targets. Three potential regulatory modules, osa-miR1432/*OsCaML2*, osa-miR1432/*OsCAS* and osa-miR1432/*OsZIP6*, were detected in our previous report [15]; however, the computational prediction of targets and expression assays of multiple time-points between three candidate regulatory modules intensively sustained the notion that *OsCaML2* may serve as a target participating in osa-miR1432-mediated rice resistance to *Xoo*. Meanwhile, though evolutionary analysis and bioinformatic prediction cannot separate *LOC_Os03g59790* from *OsCaML2* to function as a downstream actor, *OsCaML2* could withstand the stringent filtration (category ≤ 2 and raw reads ≥ 10 in at least one library) of degradome data. On this basis, the GFP-based reporter assays point to the fact that *OsCaML2* is the immune-responsive target of osa-miR1432 in rice-*Xoo* interaction. However, whether other potential targets are also implicated in osa-miR1432-mediated rice immunity to *Xoo* awaits answer in the future.

Calmodulin-like proteins (CMLs), containing EF hand motifs, were predicted to bind Ca^2+^ ions and were classified as calcium sensors [30,31]. The physiological relevance data of CMLs supported the idea that they may be involved in plant immune responses to pathogen invasion [32,33,34], and Soybean CMLs (SCaM-4/-5) and *Arabidopsis* CMLs (CML24, CML41, CML43, CML8, CML9) were confirmed to participate in the immune response of *Pseudomonas syringae* [35,36,37,38,39,40,41]. In our study, regarding the role of *OsCaML2**,* a member of the rice CML family, rice immunity response conformed to the anticipation of targets for osa-miR1432 in rice-*Xoo* interactions. Consistent with the positive immune regulator of osa-miR1432, the overexpression of *OsCaML2* exhibited a subdued resistance to *Xoo* attacks with the alteration of defense marker gene expressions, ROS levels and bacterial growth (Figure 5). Prompted by the role of the osa-miR1432/*OsCaML2* module in *Xoo* infection, the participants acting in this immune pathway were surveyed, and OsHPGT3 and OsCSN6 were identified using Y2H technology (Appendix A). Depending on the homology analysis and functional annotation, OsHPGT3 involved in arabinogalactan metabolism of the rice cell wall was the top choice to investigate the defense pathway of the osa-miR1432/*OsCaML2* module [26]. The possible interaction between OsCaML2 and OsHPGT3 revealed by Y2H and BiFC assays suggests that it relies on the EF hand domain of OsCaML2 (Appendix A). A working hypothesis was that the osa-miR1432-directed OsCaML2 repression may thicken secondary cell walls and increase lignin content by affecting the galactosyl transferase activity of OsHPGT3 protein, improving resistance to *Xoo* in rice.

## 4. Materials and Methods

### 4.1. Plant Materials and Growth Condition

Rice (*O**ryza sativa* L.) plants were cultivated in the experimental field at Changping, Beijing during the natural growing season. The *Xoo*-sensitive genotypes, IR24 and its insensitive derivative, IRBB5, served to appraise the pathogenicity of *Xoo* strain PXO86. All rice transgenic lines were generated in the background of *Japonica* rice cultivar TP309 and used for experiment in the T_2_ generation.

### 4.2. Pathogen Inoculation and Disease Assays

*Xoo* strains PXO86, PXO112, PXO99 and PXO339, preserved in a −80 °C freezer, were streaked on PSA (10 g/L tryptone, 10 g/L sucrose, 1 g/L sodium-glutamate and 10 g/L agar, pH = 7.0) plates with cephalexin (15 mg/L) and incubated at 28 °C. After 3 d, the cells harvested from the plates were resuspended in sterilized distilled water (OD_600_ = 0.5). Rice leaves were clipped using scissors with *Xoo* suspension or water at the tillering stage [42], and the lesion lengths of 15 individual plants with three repeats were measured 20 d after inoculation for each strain. 

For bacterial growth analysis, 3 infected leaves from independent lines were adequately ground utilizing a mortar and pestle. Then, the milled leaf was suspended in 10 mL of bacteria-free water, and the diluted mixture was transferred to PSA plates containing 15 mg/L of cephalexin. Finally, the cell colonies were counted from the plates incubated at 28 °C for 3 d.

### 4.3. Vector Construction and Rice Transformation

For functional overexpression, osa-miR1432 precursors and *OsCaML2* were amplified from TP309 genomic sequence and full-length complementary DNA (cDNA), respectively. The fragments were cloned into the plant binary vector, UBI-pCAMBIA1300, and then transformed into TP309 calli via *Agrobacterium* strain EHA105-mediated transformation. To create a target mimicry construct for silencing osa-miR1432 expression, we inserted the artificial target mimic of osa-miR1432 into the *IPS1* gene [23]. Then, the mutated fragments were cloned into the binary vector, pCAMBIA1300, to generate MIM1432 construct, which was introduced into TP309 calli through *Agrobacterium* strain EHA105-mediated transformation. All primers for vector construction are listed in Appendix A.

### 4.4. Quantitative Reverse Transcription-PCR Assay

For gene expression analysis, total RNA was extracted from leaves at different collected time points with or without *Xoo* treatment using TRIzol reagent (Invitrogen, Waltham, MA, USA) and reverse transcribed to cDNA using the ReverTra Ace 1 qPCR RT Master Mix with gDNA Remover (TOYOBO, Osaka, Japan). To monitor osa-miR1432, the miRNAs were extracted from the aforementioned samples using an miRNA Isolation Kit (OMEGA, Atlanta, GA, USA), and the specific stem-loop RT primer for osa-miR1432 was applied to the reverse transcription protocol [43]. Quantitative reverse transcription-PCR (qRT-PCR) was performed using indicated primers and TransStart Green qRT-PCR Super Mix (TransGen, Beijing, China). SnRNA U6 and *OsACTIN* genes separately served as the internal reference to data normalization for miRNAs and mRNAs, and the relative expression levels of miRNAs and target genes were determined using a one-way ANOVA followed by post hoc Tukey HSD analysis with three biological replicates and at least three technical repeats. Specific primers for quantitative real-time PCR are listed in Appendix A. 

### 4.5. Transient Expression Analysis in Nicotiana Benthamiana

Transient co-expression assays were carried out by injecting the *Agrobacterium* strain EHA105 mixtures, which harbored the respective expression constructs, 35S::miR1432 (pCAMBIA1300), 35S::OsCaML2 (pCAMBIA1300-221-GFP) and 35S::MIM1432 (pCAMBIA1300), into leaves of 4-week-old *N. benthamiana* plants. In brief, *Agrobacterium* strain EHA105 was shaken at 28 °C overnight in LB liquid media containing kanamycin (50 mg/mL) and rifampicin (50 mg/mL) with 250 rpm. The cells were collected at 8000 rpm for 3 min and resuspended in an MMA buffer (10 mM MES, 10 mM MgCl_2_, 200 µM acetosyringone, pH = 5.6) until OD_600_ = 1.0. After rest at temperature for 2–3 h, the *Agrobacterium* strain EHA105 containing the destination constructs were infiltrated into leaves of *N. benthamiana* for the transient expression assay. GFP accumulation was detected at 48 h post infiltration (hpi) for imaging using ZEISS LSM 710 NLO (Carl Zeiss, Oberkochen, Germany)

### 4.6. Oxidative Burst Assays

DAB and NBT staining were conducted for the determination of H_2_O_2_ and O_2_^−^ with modified protocol [44]. Briefly, 15 excised rice leaf sections from 5 independent lines were submerged in the staining solution (10 mM Tris-HCl (pH = 6.5) containing 1 mg/mL of l DAB (Sigma, St. Louis, MO, USA) or 10 mM sodium citrate (pH = 6.0) containing 0.05% NBT (Sigma, St. Louis, MO, USA) and then incubated at 37 °C in the dark for 12 h. The chlorophyll of leaves was washed out with bleaching solution (ethanol:acetic acid:glycerol = 3:1:1) in a boiling water bath (90–95 °C) until the leaves became clear. Finally, the stained leaves were photographed using an HP LJ M1530 scanner (Hewlett-Packard, Palo Alto, CA, USA).

### 4.7. Y2H Assays

Y2H assays were performed in consonance with the Matchmaker Gold Y2H System user’s manual (Clontech, Mountain View, CA, USA). The full coding sequences, truncated N and C terminal coding region of *OsCaML2,* were introduced into the pGBKT7 vector. The bait plasmid pGBKT7-*OsCaML2* was co-transformed with rice cDNA library for screening interacting proteins on SD/-Leu-Trp-selected plates, and different dilutions (10^−1^, 10^−2^ and 10^−3^) of positive clones were transferred to SD/-Trp-Leu-His-Ade medium with β-galactosidase (*X*-*α*-Gal) to confirm the interaction. After incubation at 30 °C for 3 or 4 d, yeast growth was assessed. By the same method, the coding sequence of *Os**HGPT3* was cloned into pGADT7 for specific interaction of OsCaML2 and OsHGPT3.

### 4.8. BiFC Assay

The full-length coding regions of *OsCaML2* and *OsHGPT3* were amplified and cloned into the pVYNE and pVYCE vectors, respectively. The recombinant vectors were co-transformed into 4-week-old *N. benthamiana* leaves via *Agrobacterium* strain EHA105-mediated transient expression. Briefly, the bacteria cells were collected and resuspended in infection solution (10 mM MES, 10mM MgCl_2_ and 200 μM of acetosyringone, pH = 5.6). The prepared suspensions were infiltrated into 4-week-old *N. benthamiana* leaves, and the infiltrated *N. benthamiana* plants were grown for 2 d. Fluorescent eYFP signals were detected and photographed using ZEISS LSM 710 NLO (Carl Zeiss, Oberkochen, Germany.excitation 514 nm, emission 525-565 nm). 

## 5. Conclusions

In conclusion, our study demonstrates the regulatory roles of the osa-miR1432/*OsCaML2* module in the resistance of rice against *Xoo* and displays new evidence for capturing minor resistance-associated genes by anchoring the targets of miRNAs, which may offer direction for the breeding or genetic engineering of *Xoo*-resistant rice plants.

## Figures and Tables

**Figure 1 ijms-22-11367-f001:**
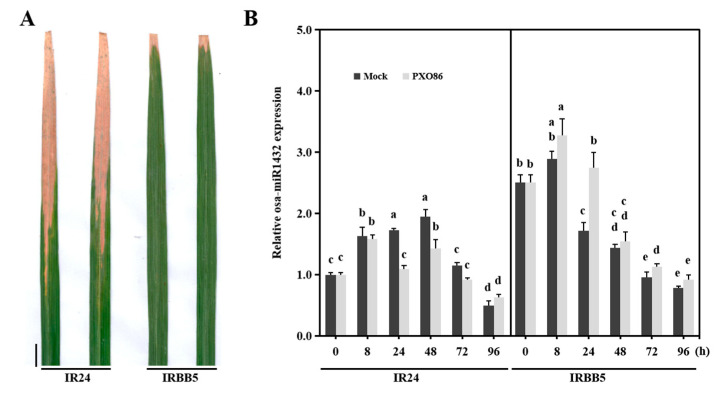
Differential expression patterns of osa-miR1432 in susceptible and resistant genotypes upon *Xoo* infection. (**A**) Phenotype of representative leaf sections from susceptible genotype IR24 and resistant genotype IRBB5 at 20 dpi with *Xoo* strain PXO86. (**B**) Accumulation of osa-miR1432 was monitored using RT-qPCR in susceptible and resistant genotypes under *Xoo* strain PXO86 and mock treatment. Scale bar, 2 cm. Error bars indicate standard deviation (*n* = 3). Different letters above the bars indicate significant differences (*p* < 0.01) determined by one-way analysis of variance (ANOVA) followed by post hoc Tukey HSD analysis.

**Figure 2 ijms-22-11367-f002:**
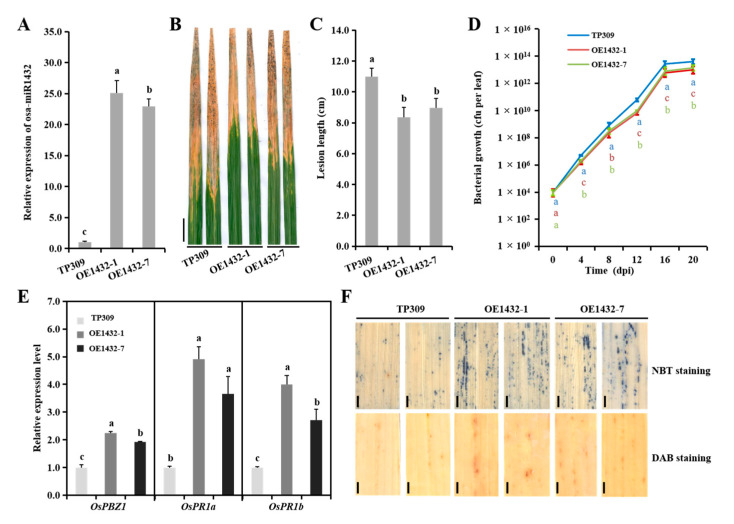
Osa-miR1432 modulates *Xoo* resistance in rice. (**A**) The expression detection of osa-miR1432 in representative overexpressing transgenic lines. SnRNA U6 served as the internal reference. (**B**) Lesions on OE1432 lines (OE1432-1 and OE1432-7) and the TP309 at 20 dpi. Scale bar, 2 cm. (**C**) Lesion lengths on OE1432 plants (*n* = 20) at 20 dpi. (**D**) Bacterial growth indicated by the numbers of colony-forming units (cfu) per leaf in OE1432 lines and TP309. (**E**) The basic expression of *OsPBZ1*, *OsPR1a* and *OsPR1b* in OE1432 lines and TP309. (**F**) The histochemical detection of ROS levels using NBT and DAB staining in OE1432 lines and TP309 at 4 dpi. Error bars indicate standard deviation (*n* = 3). Scale bar, 2 mm. Different letters above the bars indicate significant differences (*p* < 0.01) determined by one-way analysis of variance (ANOVA) followed by post hoc Tukey HSD analysis.

**Figure 3 ijms-22-11367-f003:**
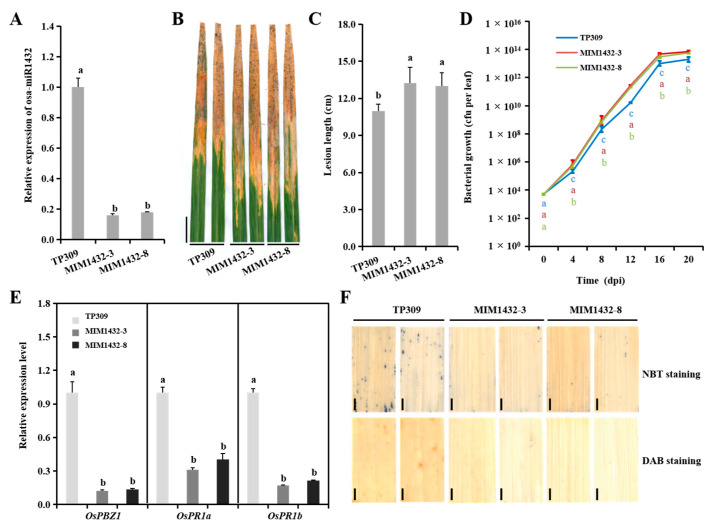
Restraining the expressing of osa-miR1432 results in susceptibility to *Xoo* attack in rice. (**A**) The expression detection of osa-miR1432 in MIM1432 lines, SnRNA U6, served as the internal reference. (**B**) Lesions on MIM1432 lines (MIM1432-3 and MIM1432-8) and TP309 at 20 dpi. Scale bar, 2 cm. (**C**) Lesion lengths on MIM1432 lines (*n* = 20) at 20 dpi. (**D**) Bacterial growth as indicated by the numbers of colony-forming units (cfu) per leaf in MIM1432 lines and TP309. (**E**) The basic expression of *OsPBZ1*, *OsPR1a* and *OsPR1b* in MIM1432 transgenic lines and TP309. (**F**) The histochemical detection of ROS levels using NBT and DAB staining in MIM1432 lines and TP309 at 4 dpi. Error bars indicate standard deviation (*n* = 3). Scale bar, 2 mm. Different letters above the bars indicate significant differences (*p* < 0.01) determined by one-way analysis of variance (ANOVA) followed by post hoc Tukey HSD analysis.

**Figure 4 ijms-22-11367-f004:**
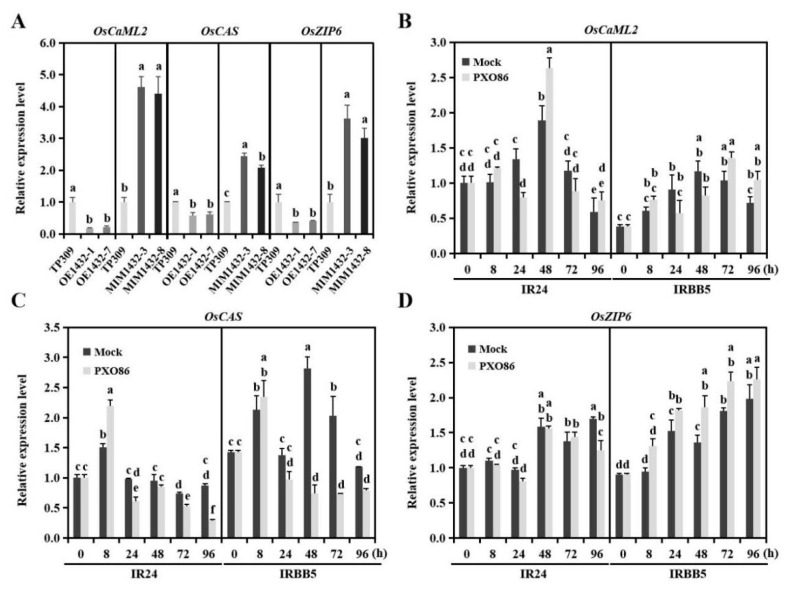
The candidate targets of osa-miR1432 have differential expression profiles in susceptible and resistant genotypes upon *Xoo* infection. (**A**) The relative expression levels of the candidate targets in the OE1432 and MIM1432 lines. (**B**–**D**) The expression pattern of *OsCaML2, OsCAS* and *OsZIP6* in susceptible and resistant genotypes upon *Xoo* or mock treatment. Error bars indicate standard deviation (*n* = 3). Different letters above the bars indicate significant differences (*p* < 0.01) determined by one-way analysis of variance (ANOVA) followed by post hoc Tukey HSD analysis.

**Figure 5 ijms-22-11367-f005:**
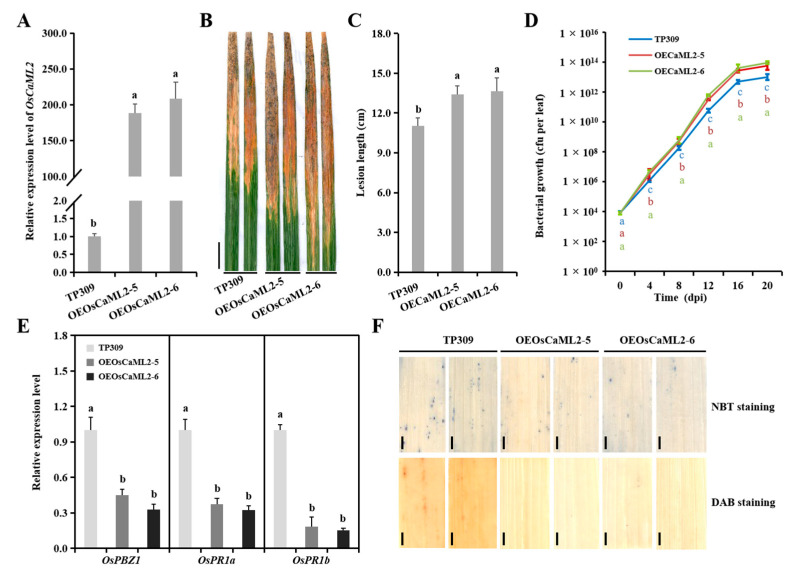
*OsCaML2* negatively regulates disease resistance against *Xoo* in rice. (**A**) Accumulation of *OsCaML2* in the OEOsCaML2 transgenic lines. (**B**) Lesions on respective OEOsCaML2 lines (OEOsCaML2-5 and OEOsCaML2-6) and the TP309 at 20 dpi. Scale bar, 2 cm. (**C**) Lesion lengths on OEOsCaML2 plants (*n* = 20) at 20 dpi. (**D**) Bacterial growth as indicated by the numbers of colony-forming units (cfu) per leaf for OEOsCaML2 lines and the TP309. (**E**) Expression pattern of *OsPBZ1*, *OsPR1a* and *OsPR1b* in OEOsCaML2 transgenic lines and the TP309. (**F**) The histochemical detection of ROS levels using NBT and DAB staining in OEOsCaML2 transgenic lines and the TP309 after *Xoo* infection. Error bars indicate SD (*n* = 3). Scale bar, 2 mm. Different letters above the bars indicate significant differences (*p* < 0.01) determined by one-way analysis of variance (ANOVA) followed by post hoc Tukey HSD analysis.

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
