# Peer review of "Inducible Enrichment of Osa-miR1432 Confers Rice Bacterial Blight Resistance through Suppressing OsCaML2"

_ijms, 2021, doi:10.3390/ijms222111367_

Round 1
Reviewer 1 Report
The manuscript describes the identification of a specific OsmiR1432 conferring bacterial blight resistance in a resistant genotype of rice compared to a susceptible one. Overexpression and knock-down lines of this miRNA were investigated and target genes identified. Expression of disease marker genes was measured with qRT-PCR and a target gene was confirmed with GFP-based reporter assays. Finally, an OsmiR1432/OsCaML2 module was postulated as important for Xoo resistance.
The results are interesting but main questions remain.
If higher expression of OsmiR1432 improved the resistance of plants against bacterial blight resistance, why this is accompanied with a higher ROS accumulation, usually a sign for larger damage. In connection to that, it was not reported how many leaves were investigated for the ROS measurements. An image analysis of the ROS staining would be desirable to quantify the results and to enable a significance analysis.
It is absolutely necessary to explain the results shown in Figure S6 and S7 in detail in the results section and not only mention them shortly in the discussion (L247).
Furthermore, it should be explained in detail how the three targets were chosen (L151-154), e.g. in the method section. This applies also to line 241, where a stringent filtration of degradome data was mentioned, but not explained in detail.
The influence of higher expression of OsmiR1432 on growth should be discussed in more detail with the knowledge that OsmiR1432 also functions in growth and development as the authors claim themselves (L221).
CaML proteins were previously classified as calcium sensor proteins. This is not discussed at all in the discussion and needs detailed consideration. Instead, authors claim that relevant functions of these proteins have not been reported yet (L186).
The conclusion that Osa-miR1432 may adopt SA-JA/ET signaling pathways as priming strategy against biotic invaders (L227) is only based on higher expression of three single genes and is very hypothetical without any further evidences. This sentence should be left out or should be rephrased.
The language throughout the whole manuscript has to be intensively improved, several sentences are difficult to understand.
Minor:
L27: with plant
L36: Multivarious
L58-50: check sentence
L68: Osa-miR1432 expression is induced in resistant genotypes during Xoo infection
Fig. 2D: Significances are missing or hard to see
L161: no correlation analysis was done, avoid the term “correlation”, whole sentence not clear
Conclusion: Should be rephrased for better understanding
Author Response
The manuscript describes the identification of a specific OsmiR1432 conferring bacterial blight resistance in a resistant genotype of rice compared to a susceptible one. Overexpression and knock-down lines of this miRNA were investigated and target genes identified. Expression of disease marker genes was measured with qRT-PCR and a target gene was confirmed with GFP-based reporter assays. Finally, an OsmiR1432/OsCaML2 module was postulated as important for Xoo resistance.
The results are interesting but main questions remain.
- If higher expression of OsmiR1432 improved the resistance of plants against bacterial blight resistance, why this is accompanied with a higher ROS accumulation, usually a sign for larger damage.In connection to that, it was not reported how many leaves were investigated for the ROS measurements. An image analysis of the ROS staining would be desirable to quantify the results and to enable a significance analysis.
Reply: Thanks for your positive comment, as suggested, we have added the relevant information on the higher ROS accumulation and leaves for ROS measurements in the updated manuscript (please see page 3 line 113 to line 114 and page 10 line 363). Meanwhile, we think that some explanations should be done for better understanding. As for the higher ROS accumulation, previous reports have shown that plant answers the pathogen invasion and multiplication through disease resistance and cell death (Wang et al., 2020; Jacob et al., 2021), our work displayed that the osa-miR1432-mediated resistance dependents on the inhibition of bacterial growth and the cell death induced by higher ROS accumulation. On the other hand, it is widely accepted to utilize qualitative experiments to detect ROS production via NBT and DAB staining method in plant-pathogen interaction (Song et al., 2021; Wang et al., 2018; You et al., 2018) and the data for image analysis of the ROS staining was not available and effective to quantify by using image J software.
- It is absolutely necessary to explain the results shown in Figure S6 and S7in detail in the results section and not only mention them shortly in the discussion (L247).
Reply: Thanks for your positive comment, as suggested, we have moved the relevant information on the results shown in Figure S6 and S7 to the results section in the updated manuscript (please see page 7 line 213 to 230).
- Furthermore, it should be explained in detail how the three targets were chosen (L151-154), e.g. in the method section. This applies also to line 241, where a stringent filtration of degradome data was mentioned, but not explained in detail.
Reply: Thanks for your positive comment, as suggested, we have added the screening criteria and relevant description for the targets in the updated manuscript (please see page 5 line 162 to line 167 and page 8 line 278 to line 279).
- The influence of higher expression of OsmiR1432 on growth should be discussed in more detail with the knowledge that OsmiR1432 also functions in growth and development as the authors claim themselves (L221).
Reply: Thanks for your positive comment, as suggested, we have added the relevant information for osa-miR1432 with higher expression on growth and development in the updated manuscript (please see page 8 line 255 to line 260 ).
- CaML proteins were previously classified as calcium sensor proteins. This is not discussed at all in the discussion and needs detailed consideration. Instead, authors claim that relevant functions of these proteins have not been reported yet (L186).
Reply: Thanks for your positive comment, as suggested, we have added the relevant information for CaML proteins in the updated manuscript (please see page 8 line 283 to line 288). Additionally, the sentence “the relevant function studies of these proteins have not been reported.” in the Line 193-194 was used to describe the notion that the exact biological function report on the homologues of OsCML2, not all CaML proteins, have not been reported, and has revised to “the relevant biological function studies of these 38 proteins have not been reported.”.
- The language throughout the whole manuscript has to be intensively improved, several sentences are difficult to understand.
Reply: Thanks for your positive comment, as suggested, we have rewritten the puzzling sentences distributed in the whole manuscript and added some explanations to be better understood in the updated manuscript (please see page 1 line14 to line 15 and line 18 to line 20, page 2 line 56 to line 59 and line 61 to line 64, page 3 line 109 to line 111 and line 116 to line 117, page 4 line 135, page 6 line 211, page 8 line 255 to line 260 and page 8 line 273 to line 274, etc).
Minor:
L27: with plant
Reply: Thanks for your positive comment, as suggested, we have revised the sentence in the updated manuscript (please see page 1 line 27 to line 29).
L36: Multivarious
Reply: Thanks for your positive comment, as suggested, we have corrected the word in the updated manuscript (please see page 1 line 37).
L58-50: check sentence
Reply: Thanks for your positive comment, as suggested, we have revised the sentence in the updated manuscript (please see page 2 line 51 to line 64).
L68: Osa-miR1432 expression is induced in resistant genotypes during Xoo infection
Reply: Thanks for your positive comment, as suggested, we have revised the sentence in the updated manuscript (please see page 2 line 72).
Fig. 2D: Significances are missing or hard to see
Reply: Thanks for your positive comment, as suggested, we have added the relevant information into Figure 2D in the updated manuscript (please see in Figure 2D, Figure 3D, Figure 5D).
L161: no correlation analysis was done, avoid the term “correlation”, whole sentence not clear
Reply: Thanks for your positive comment, as suggested, we have avoided the term with the appropriate descriptive words or sentences in the updated manuscript (please see page 5 line173 to line 175).
Conclusion: Should be rephrased for better understanding
Reply: Thanks for your positive comment, as suggested, we have rewritten the conclusion to be better understood in the updated manuscript (please see page 10 line 391 to line 395).
References
Jacob, P.; Kim, N.H.; Wu, F.H .; El-Kasmi, F.; Chi, Y.; Walton, W.G.; Furzer, O.J.; Lietzan, A.D.; Sunil, S.; Kempthorn, K.; et al. Plant "helper" immune receptors are Ca2+-permeable nonselective cation channels. Science 2021, 373, 420-425.
You, J.; Fang, Y. J.; Xiong, L.Z. Reactive oxygen detection. Bio-101 e1010170. 2018, Doi: 10.21769/BioProtoc.1010170.
Wang, Z.Y.; Xia, Y.Q.; Lin, S.Y.; Wang, Y.R.; Guo, B.H.; Song, X.N.; Ding, S.C.; Zheng, L.Y.; Feng, R.Y.; Chen, S.L.; et al. Osa-miR164a targets OsNAC60 and negatively regulates rice immunity against the blast fungus Magnaporthe oryzae. Plant J. 2018, 95, 584-597.
Wang, W.; Feng, B.; Zhou, J.M.; D.Z. Plant immune signaling: Advancing on two frontiers. J. Integr. Plant Biol. 2020, 62, 2-24.
Song, Y.; Wilson, A.J.; Zhang, X.C.; Thoms, D.; Sohrabi, R.; Song, S.Y.; Geissmann, Q.; Liu, Y.; Walgren, L.; He, S.Y.; et al. FERONIA restricts Pseudomonas in the rhizosphere microbiome via regulation of reactive oxygen species. Nat. Plants 2021, 7, 644-654.
Reviewer 2 Report
This manuscript is well written and provided the novel outcome for improving the Xanthomonas resistance on rice crop. The outcome of the study is indicating that Osa- 17 miR1432/OsCaML2 expressed differentially in response to Xanthomonas infection. All the sections were written concisely and in a reproducible manner, however, there are some typos and different fonts in the manuscript. Besides, material and methods have to be rechecked to ensure that they were replicated from their previously published articles. I highly recommend this article for publication
Some typo checks and fonts (e,g. Line 19, Line 102, Line 319-328)
Author Response
This manuscript is well written and provided the novel outcome for improving the Xanthomonas resistance on rice crop. The outcome of the study is indicating that Osa-miR1432/OsCaML2 expressed differentially in response to Xanthomonas infection. All the sections were written concisely and in a reproducible manner, however, there are some typos and different fonts in the manuscript. Besides, material and methods have to be rechecked to ensure that they were replicated from their previously published articles. I highly recommend this article for publication
Some typo checks and fonts (e,g. Line 19, Line 102, Line 319-328)
Reply: Thanks for your positive comment, as suggested, we have corrected the typos and fonts and revised materials and methods in the updated manuscript (please see page 1 line 20, page 3 line 95, line 106 and line 119, page 8 line 261, line 294 and line 298, page 9 line 300, line 308, line 312 to line 313, line 328 and line 330, page 10 line 354, line 356, line 367 to line 368, line 377 to line 378 and line 384 to line 386, etc).
Round 2
Reviewer 1 Report
The authors carefully considered the comments of the referee and improved the manuscript according to the suggestions of the reviewer. Most changes contribute to a better understanding of the discussed topic. The revised manuscript is now ready for publishing after small corrections mentioned below.
L45-47, L49: resistant genes and susceptible genes do not exist, please change either to genes conferring resistance or resistance and susceptibility genes
L114: more severe
L113/114: I still do not understand why disease resistance should be accompanied by more severe cell damage?
L114-117: If growth rates of Xoo strain PXO86 were lower in OE1432 transgenic rice plants as shown in Figure 2D), then this implies that osa-miR1432-mediated resistance may act in the inhibition of Xoo strain multiplication.
L173-175: sentence not clear at all. There are no transcriptional time-points of OsCaML2 to osa-miR1432. Please delete or rephrase to transcription of …. at certain time points.
L272: cite the previous report here!!!
L282: rice immunity
L289: overexpressors
L289-292: check long sentence for clarity. The sentence should be divided into two for better understanding.
L393: resistance-associated
Author Response
The authors carefully considered the comments of the referee and improved the manuscript according to the suggestions of the reviewer. Most changes contribute to a better understanding of the discussed topic. The revised manuscript is now ready for publishing after small corrections mentioned below.
L45-47, L49: resistant genes and susceptible genes do not exist, please change either to genes conferring resistance or resistance and susceptibility genes
Reply: Thanks for your positive comment, as suggested, we have corrected the word in the updated manuscript (please see page 1 line 21, line 27 and line 42, page 2 line 45 to line 46 and line 49).
L114: more severe
Reply: Thanks for your positive comment, as suggested, we have added the word in the updated manuscript (please see page 3 line 113 to line 114).
L113/114: I still do not understand why disease resistance should be accompanied by more severe cell damage?
Reply: Thanks for your positive comment, we think that additional explanations should be as detailed and clear as possible for better understanding. Previous reports have shown that the early recognition event determines plant disease resistance to pathogens during the infection process, which is known as "hypersensitive response" (HR), leading to a rapid tissue cell damage or necrosis at the site of infection and providing resistance to the great majority of potential pathogens (Keen et al., 1990; Mehdy et al., 1994). As for the ROS, the characteristic of HR is rapid generation of O2·- and accumulation of H2O2 (Doke et al., 1983; Lamb et al., 1997), therefore, it is widely accepted that the production of ROS serves as typical immune response indicators and disease resistance should be accompanied by more severe cell damage at the site of infection. Our work also displayed that the osa-miR1432-mediated resistance relies on the more severe cell damage, even cell death, induced by higher ROS accumulation.
L114-117: If growth rates of Xoo strain PXO86 were lower in OE1432 transgenic rice plants as shown in Figure 2D), then this implies that osa-miR1432-mediated resistance may act in the inhibition of Xoo strain multiplication.
Reply: Thanks for your positive comment, as suggested, we have revised the sentence in the updated manuscript (please see page 3 line 117).
L173-175: sentence not clear at all. There are no transcriptional time-points of OsCaML2 to osa-miR1432. Please delete or rephrase to transcription of …. at certain time points.
Thanks for your positive comment, as suggested, we have rewritten the sentences in the updated manuscript (please see page 5 line 172 and line 175 to line 177).
L272: cite the previous report here!!!
Reply: Thanks for your positive comment, as suggested, we have added the relevant reference in the updated manuscript (please see page 8 line 274).
L282: rice immunity
Reply: Thanks for your positive comment, as suggested, we have corrected the word in the updated manuscript (please see page 8 line 284 and 291).
L289: overexpressors
Reply: Thanks for your positive comment, as suggested, we have corrected the word in the updated manuscript (please see page 8 line 293).
L289-292: check long sentence for clarity. The sentence should be divided into two for better understanding.
Reply: Thanks for your positive comment, as suggested, we have revised the sentence in the updated manuscript (please see page 8 line 291 to 293).
L393: resistance-associated
Thanks for your positive comment, as suggested, we have corrected the word in the updated manuscript (please see page 11 line 396).
References
Doke, N. Involvement of superoxide anion generation in the hypersensitive response of potato tuber tissues to infection with an incompatible race of Phytophthora infestans and to the hyphal wall components. Physiol. Plant Pathol. 1983, 23, 345-357.
Keen, N.T. Gene-for-gene complementarity in plant-pathogen interactions. Annu. Rev. Genet. 1990, 24, 447-463.
Lamb, C.; Dixon, R.A. The oxidative in plant disease resistance. Annu. Rev. Plant Physiol. Plant Mol. Biol. 1997, 48, 251-275.
Mehdy, M.C. Active oxygen species in plant defense against pathogens. Plant Physiol. 1994, 105, 467-472.